# Genome Characterization of Bacteriophage KPP-1, a Novel Member in the Subfamily *Vequintavirinae*, and Use of Its Endolysin for the Lysis of Multidrug-Resistant *Klebsiella variicola* In Vitro

**DOI:** 10.3390/microorganisms11010207

**Published:** 2023-01-13

**Authors:** Amal Senevirathne, Jehee Lee, Mahanama De Zoysa, Chamilani Nikapitiya

**Affiliations:** 1College of Veterinary Medicine and Research Institute of Veterinary Medicine, Chungnam National University, Daejeon 34134, Republic of Korea; 2Fish Vaccine Research Center, Jeju National University, Jeju 63243, Republic of Korea

**Keywords:** *Klebsiella pneumoniae* complex, bacteriophage, endolysin, genome sequence, *Klebsiella variicola*, *Mydovirus*, multi-drug resistance

## Abstract

Multidrug-resistant members of the *Klebsiella pneumoniae* complex have become a threat to human lives and animals, including aquatic animals, owing to the limited choice of antimicrobial treatments. Bacteriophages are effective natural tools available to fight against multidrug-resistant bacteria. The bacteriophage KPP-1 was found to be strictly lytic against *K. variicola*, a multidrug-resistant isolate, producing clear plaques. The genome sequence analysis of KPP-1 revealed that it comprised 143,369 base pairs with 47% overall GC content. A total of 272 genes (forward 161, complementary 111) encode for 17 tRNAs and 255 open reading frames (ORFs). Among them, 32 ORFs could be functionally annotated using the National Center for Biotechnology Information (NCBI) Protein Basic Local Alignment Search Tool (BLASTp) algorithm while 223 were found to code for hypothetical proteins. Comparative genomic analysis revealed that the closest neighbor of KPP-1 can be found in the genus *Mydovirus* of the subfamily *Vequintavirinae*. KPP-1 not only markedly suppressed the growth of the host but also worked synergistically with ampicillin. Useful genes for pathogen control such as endolysin (locus tag: KPP_11591) were found to have activity against multidrug-resistant isolate of *K. variicola*. Further studies are necessary to develop a strategy to control the emerging pathogen *K. variicola* using bacteriophages such as KPP-1.

## 1. Introduction

The *Klebsiella pneumoniae* complex consists of seven *K. pneumoniae*-related species including *K. variicola*. It is a versatile bacterium infecting a variety of hosts, such as humans, animals, insects, and plants [1]. The virulence profile of *K. variicola* has not been fully characterized. Its clinical significance is obscured because of its incorrect identification as other members of the *K. pneumoniae* complex. The biochemical and phenotypic properties of members of the *K. pneumoniae* complex often overlap with each other; thus, traditional microbiological methods are insufficient for correct classification [2,3]. *Klebsiella variicola* is known to possess plasmid-acquired multidrug resistance [4]. Therefore, it is a medically important microorganism that causes difficult-to-treat infections. With numerous reports of *K. variicola* infections in humans worldwide, it is now being considered an emerging pathogen [5,6,7]. According to some incidences of bloodstream infections, the virulence of *K. variicola* is even higher than that of *K. pneumoniae* [8]. Furthermore, *K. variicola* is known to cause urinary tract infections (UTIs) and sepsis in humans [1]. Additionally, there is evidence of community-acquired post-surgical meningitis caused by *K. variicola* [9]. The clinical landscape of the bacterium is changing at a rapid pace. Thus, the bacterium has drawn significant attention among the medical and research communities. Furthermore, infections of *K. variicola* are more common among humans than among farm animals and aquatic resources. Owing to its increasing importance, measures to control *K. variicola* in human environments are essential, preferably without employing conventional antibiotics. In this context, use of bacteriophages can be one of the most effective and innovative approaches, which does not involve the risk of the development of antibiotic resistance while being a completely eco-friendly way of controlling the spread of pathogens.

Bacteriophages, generally termed phages are viruses that specifically recognize and infect their host bacterium. The fate of the host bacterium is decided depending on the lifestyle of the bacteriophage which may be lytic or lysogenic. Over a century of research on bacteriophages has completely changed the landscape of basic biology and medicine. Among a multitude of uses of bacteriophage genetic resources in biotechnology, phage therapy has gained enormous popularity for successfully controlling multidrug-resistant pathogens during a post-antibiotic era, where the majority of frontline antibiotics have failed to confer the intended protection [10,11]. There are reports of successful use of phage therapy in humans to treat lethal diseases such as pneumonia and tuberculosis [12,13,14], in veterinary fields to control cow mastitis by utilizing cocktails of bacteriophages [15], and in industrial fields, such as food, agriculture, and experimental aquaculture conditions [16].

To date, two reference genomes, and data on 23 annotated *Klebsiella*-infecting phages are available in the National Center for Biotechnology Information (NCBI) database [17]. These 23 *Klebsiella* phages belong to 13 genera. The enormous wealth of phage genomic resources can be explored to identify proteins that can be utilized in pathogen control. However, one major limitation of using bacteriophage genetic resources is that the majority of proteins remain putative candidates that are yet to be fully characterized. Some of the well-characterized phage-related proteins are endolysins, the enzyme which plays a key role in bacterial lysis [18]. Owing to the presence of distinct cell wall binding and catalytic domains, endolysins have been experimentally utilized for the control of pathogenic bacteria and their detection via various biosensor platforms [19]. The key virulence factor of *K. pneumoniae* complex pathogens is the polysaccharide capsule. Hence, enzymes that can damage these structures are therapeutically important to develop novel bacterial control strategies. Furthermore, by economizing strict specificity to the host bacterium, bacteriophages can be utilized for bacterial classification and identification using a strategy generally known as phage typing. These potential avenues can be examined with respect to the KPP-1 bacteriophage in further studies.

In the present study, we introduced a novel, strictly lytic bacteriophage to the multidrug-resistant *K. variicola* isolate and conducted genomic and phenotypic characterization of the bacteriophage. Utilizing electron microscopy data, genome sequence data, and proteomic data, we conducted phylogenetic classification and proposed the novel *K. variicola* phage to be a member of the genus *Mydovirus*, of the subfamily *Vequintavirinae* of the class *Caudoviricetes*. Due to the absence of toxin-proteins, integrases, or antibiotic resistance-related proteins, KPP-1 is a potential candidate for therapeutic applications, which require further studies on the phage. Its ability to cause rapid bacterial lysis and capacity to be utilized along with antibiotics has channeled our studies into a novel direction whether phages can augment the function of antibiotics. Therefore, further studies are necessary to utilize the novel bacteriophage KPP-1 for the betterment of humanity by controlling yet another lethal pathogen *K. variicola* in human environments.

## 2. Materials and Methods

### 2.1. Bacterial Strains and Culture Conditions

The multidrug-resistant *K. variicola* strain used for bacteriophage isolation was a laboratory isolate from fresh stream water. Routine cultures of the *Klebsiella* strain were maintained in nutrient broth (NB; BD, Sparks, NV, USA) at 37 °C with vigorous shaking. *Cronobacter sakazakii*, *Enterococcus faecalis*, *Escherichia coli*, *Enterobacter cloacae*, and *Shigella flexneri* isolates were grown in Luria Bertani broth (LB; BD, Sparks, NV, USA), *Morganella morganii*, and *Pantoea dispersa* were grown in NB, all *Edwardsiella piscicida* isolates were grown in brain heart infusion broth (BHI, BD, Sparks, NV, USA) supplemented with 1% NaCl (BHI + 1% NaCl). *Aeromonas hydrophila*, *A. salmonicida*, *A. sobria*, *A. veronii*, *Providencia rettgeri*, *Staphylococcus aureus*, *S. haemolyticus*, and *Listeria monocytogenes* isolates were grown in tryptic soy broth (TSB, BD, Sparks, NV, USA). *Vibrio crassostrea*, *V. cyclitrophicus*, *V. ichthyoenteri*, *V. splendidus*, and *V. tubiashi* isolates were grown in BHI + 1% NaCl broth. Species-level identity of each bacterial strain was confirmed at the molecular level by sequencing 16S rRNA using universal primers 27F and 1492R (Cosmo Genetech Inc., Daejeon, Republic of Korea). Sequence alignment and comparison of aligned sequence fractions against public databases were performed using NCBI, Basic Local Alignment Search Tool-nucleotide suite (NCBI BLASTn tool), and using the EzTaxon-e database [20,21].

### 2.2. Isolation of Bacteriophage KPP-1 That Infects K. variicola

Water samples collected from the Gapcheon River, Daejeon, Republic of Korea were enriched with the *K. variicola* isolate which was used as a propagation strain. Briefly, 10 mL of water sample was diluted into 90 mL NB supplemented with 20 mM CaCl_2_ and MgCl_2_ and incubated (at 37 °C) overnight in 500 mL conical flasks with vigorous shaking (180 rpm). After incubation, enriched samples were subjected to centrifugation at 12,300× *g* for 10 min. The supernatant was collected and filter sterilized by allowing it to pass through 0.2 μm membrane filters (Minisart, Sartorius, Goettingen, Germany). *K. variicola*-specific bacteriophages were isolated from the filtered supernatant. Ten-fold serial dilution of the sample was prepared in SM buffer (5 mM Tris-HCl, 100 mM NaCl, 10 mM MgSO_4_; pH 7.4), and 100 µL of each dilution was mixed with overnight-grown *K. variicola* isolate (indicator strain) in 4 mL of TA soft agar (0.4% agar, 8 g NB, 5 g NaCl [86 mM], 0.2 g MgSO_4_·7H_2_O [0.8 mM], 0.05 g MnSO4 [0.3 mM], and 0.15 g CaCl_2_ [1 mM] per 1 L distilled water; pH 6.0). The mixture was then overlayed onto pre-solidified NB agar plates (1.5% agar). The plates were kept still for 15 min for solidification and incubated at 37 °C for 12 h. After incubation, single clear plaques were picked using pipette tips and suspended in 500 μL of SM buffer. The suspension was filter sterilized using 0.2 μm Minisart membrane filters. Agar overlaying and single plaque isolation was conducted at least 3 consecutive times to purify newly isolated bacteriophages and the novel bacteriophage isolate was designated as KPP-1.

### 2.3. High Titer Bacteriophage Preparation

High titer bacteriophage preparation was conducted in liquid culture. Polyethylene glycol (PEG) precipitation and CsCl gradient ultracentrifugation of bacteriophage KPP-1 were performed according to methods followed in a previous report [22]. Titer of the phage preparation was determined using the plaque assay [23].

### 2.4. Host Range Analysis

Host range analysis was performed using the plaque assay on TA soft agar [24]. It included *K. variicola* (2 isolates), *K. pneumoniae* (2 isolates), *A. hydrophila* (1 isolate), *A. salmonicida* (1 isolate), *A. sobria* (1 isolate), *A. veronii* (2 isolates), *C. sakazakii* ATCC 29544, *E. coli* O157:H7 ATCC35150, *E. faecalis* (1 isolate), *E. cloacae* (1 isolate), *E. piscicida* (2 isolates), *L. monocytogenes* (1 isolate), *M. morganii* (1 isolate), *P. dispersa* (1 isolate), *P. rettgeri* (1 isolate), *S. flexneri* (2 isolates), *S. aureus* (1 isolate), *S. haemolyticus* (1 isolate), *V. crassostrea* (2 isolates), *Vibrio cyclitrophicus* (1 isolate), *V. ichthyoenteri* (2 isolates), *V. splendidus* (2 isolates), and *V. tubiashi* (2 isolates). Lysis was confirmed by the appearance of distinct plaques on the bacterial lawns after incubation for 12 to 15 h. Efficacy of infection (EOI) of KPP-1 on *K. variicola* propagation strain and a laboratory strain was conducted using dot assay. Briefly, overnight cultures of two *K. variicola* strains were prepared. One hundred microliters of bacterial culture was inoculated into 4 mL of soft TA and overlayed on pre-solidified NB plates. Plates were kept for 10 min for solidification. Serially diluted KPP-1 phage stock (1 × 10^10^ pfu/mL) was dotted on each plate (10 µL) and allowed to stand still for 20 min for complete adsorption. Plates were incubated for 12–15 h and the lysis efficacy was observed.

### 2.5. Transmission Electron Microscopy (TEM)

The TEM analysis of phage particles was conducted using purified phage particles by CsCl gradient centrifugation [25]. Ten microliters of dialyzed high titer KPP-1 phage sample (>10^11^ pfu/mL) was diluted ten-fold and was spotted on a carbon-coated copper grid and incubated for 10 min at room temperature. Excess phage suspension was removed by blotting into triangular filter paper tips. Then the phage-adsorbed carbon-coated copper grids were negatively stained with 10 μL of UA-Zero EM stain (Agar Scientific, Stansted, Essex, UK). Samples were observed using spherical aberration corrected scanning transmission electron microscope (Cs-corrected STEM (JEM-ARM200F; JEOL, Tokyo, Japan). The acceleration voltage used was 100 kV, at 100,000× magnification. At least five selected complete phages were used for measurements using the Image-J software [26].

### 2.6. Genomic DNA Isolation and KPP-1 Phage Genome Sequencing

Genomic DNA was isolated using PEG-precipitated bacteriophage stock. The bacteriophage sample was filter sterilized and DNA was isolated using GeneAll^®^ Exgene™ Cell SV DNA extraction kit (GeneAll Biotechnology Co., Ltd., Daejeon, Republic of Korea). The quality of genomic DNA was evaluated using 1% agarose gels (Appendix A). Whole genome sequencing was performed using the MiSeq sequencing system (Illumina, San Diego, CA, USA) at Macrogen Inc. (Macrogen, Geumcheon-gu, Republic of Korea). Sequence quality control and trimming were performed using FastQC V.0.11.5 [27], and Trimmomatic V.0.36 [28]. After de novo assembly of the genome using the SPAdes software V.3.15.4 [29], the resultant single contig was utilized for further bioinformatics analysis.

### 2.7. Functional Annotation

Prediction of open reading frames (ORFs) was performed using the Prokka pipeline V.1.12 [30] In addition, the whole genome was re-assessed using the Rapid Annotation using Subsystems Technology (RAST 2.0) [31], Glimmer V.3.02 [32], and GeneMark V.4.28 [33] servers and finally verified using the NCBI ORF finder tool [34]. Final gene allocations were performed by considering results predicted by all predication tools. Functional analysis and assignment and protein functions were determined using the NCBI protein Basic Local Alignment Search Tool (BLAST-p; NCBI, Bethesda, MD, USA) [20]. The prediction of tRNA sequences was performed using the ARAGORN V.1.2.36 [35] server and the tRNAscan-SE V 2.0 [36] server. Finally, the genome map view was generated using the CGView Server [37].

### 2.8. Comparative Genomic Analysis and Phylogenetic Positioning

Assessment of genome sequences for repeated sequences was conducted using the Geneious Prime software and PhageTerm python program [38,39]. Identification of the genome packaging method was conducted using the PhageTerm program by utilizing the next-generation sequencing derived fastq reads and the complete-assembled KPP-1 genome sequence (Fasta format) [39]. The genome nucleotide sequence was compared with the NCBInr/nt database using the BLASTn algorithm to identify the closest relative among available phage genomes. The DNA sequence identity was calculated by multiplying the sequence coverage by percent identity (query cover × % identity). The DNA sequence identity of KPP-1 was compared against that of classified related viruses listed by the International Committee on Taxonomy of Viruses (ICTV) [40]. The intergenomic similarity was compared using the VIRIDIC tool [41] using all the classified phages in the subfamily *Vequintavirinae* that included the members of genera *Avunavirus*, *Certrevirus*, *Henunavirus*, *Mydovirus*, *Seunavirus*, and *Vequintavirus*. Intergenomic comparison by visualization of BLAST scores was generated by Easyfig (V.2.2.5) analysis using default bootstrap values [42]. The whole genome sequence of the KPP-1 was compared against genome sequences of 129 bacteriophages (Appendix A) known to infect *Klebsiella*, *Bacillus*, *Listeria*, *Staphylococcus*, *Salmonella*, and *Escherichia* species using the CLANS software (executed for 10,000 cycles) [43]. A phylogenetic tree was constructed using capsid protein and terminase large subunit protein of members belonging to subfamily *Vequintavirinae*. The maximum likelihood tree was generated using Phylogeny.fr program One Click mode with gblock enabled [44]. A comprehensive proteomic tree of viral genomes and KPP-1 was generated using the ViPTree server V.3.2 [45].

### 2.9. Synergistic Inhibition of Bacterial Growth by KPP-1 and Antibiotics

Inhibition of *K. variicola* growth along with KPP-1, and KPP-1 in combination with ampicillin antibiotic as a model cell wall biosynthesis was assessed in vitro. The bacterium was grown overnight and diluted to formulate a fresh 1% inoculation in NB. Treatment conditions were set as control (bacteria alone), bacteria with ampicillin (100 μg/mL), bacteria with KPP-1 (10 MOI; multiplicity of infection), and bacteria with ampicillin (100 μg/mL) + KPP-1 (10 MOI). Increase in bacterial population was monitored using optical density at 595 nm (OD_595_) measurements at 0, 3, 6, 9, and 12 h post-inoculation.

### 2.10. Identification and Cloning of Endolysin

Based on the functional annotation study, the gene (locus tag KPP_11591) was recognized as the potential gene for bacterial lysis. The ORF of potential endolysin gene KPP_11591 was amplified using the polymerase chain reaction (PCR) using phage genomic DNA. The primer sequences used to amplify the KPP-1 endolysin ORF are as follows: forward primer 5′–GAGAGAGAATTCTTGAAACTTACGCTGGAACAA–3′ and reverse primer 5′–GAGAGACTCGAGTTAAGAGGTTAGAACAGATTTTGC–3′. The PCR conditions used were 94 °C for 2 min, 94 °C for 30 s, annealing temperature at 60 °C for 45 s, and elongation at 72 °C for 1 min. The amplified ORF was flanked by EcoR1 and XhoI restriction enzyme sites at the 5′ end and 3′ end, respectively, for cloning. The double-digested insert was cloned into the HisParallel 1 (6 × His tag) vector, which was digested with the same restriction enzymes. The recombinant vector was transformed into the *E. coli* DH5α strain. Resultant colonies were verified as positive using colony PCR, restriction enzyme digestion, and sequencing. Positive colonies with the correct insert were prepared for plasmid isolation. The recombinant plasmid was then transformed into *E. coli* BL21 (DE3) for overexpression and protein purification. Bacterial induction for protein purification was conducted following the standard procedure by inducing with 1 mM isopropyl β-D-a-thiogalactopyranoside (IPTG) and incubating for 10 h at 30 °C. Proteins were purified using the Ni-NTA column chromatography method (His.Bind Purification Kit, EMD Biosciences, Inc., Merck KGaA, Darmstadt, Germany). The purity of overexpressed protein was confirmed using 12% sodium dodecyl-sulfate polyacrylamide gel electrophoresis (SDS-PAGE) [46] and protein concentration was measured using the Bradford method [47]. The structural model of the endolysin was generated using SWISS-MODEL server [48]. Each domain was identified using the InterProScan search tool [49].

### 2.11. Determination of the Effect of KPP-1 Endolysin on Bacterial Cells Using Live and Dead Cell Assay

The effect of endolysin on bacterial membrane permeability was assessed using live and dead cell assay using the propidium iodide (PI; Sigma, St. Louis, MO, USA) staining method. To identify the live bacterial cells, fluorescein diacetate (FDA; Sigma, St. Louis, MO, USA) staining was used. Bacterial cultures were grown until the early-log phase (0.18 OD_595_). Cells were collected by centrifugation at 9800× *g* for 10 min using a tabletop centrifuge (Smart R17, Hanil, Kimpo, Republic of Korea). Cells were re-suspended in NB and treated with 350 µg/mL KPP-1 endolysin for 10 h. Cells were collected by centrifugation and washed with 1× phosphate buffered saline (PBS) two times. Then cells were treated with PI (5 µg/mL) and FDA (5 µg/mL) for 25 min at 25 °C on ice in darkness. Samples were quenched by re-suspending in PBS. Cells prepared in appropriate dilution were placed on glass slides and covered with coverslips. Cells were observed using confocal laser scanning microscopy (CLASM) using a scan head integrated with an Axiovert 200 M inverted microscope (Carl Zeiss, Jena, Germany). For imaging, the excitation and emission wavelength for red fluorescence (PI) were 535 and 617 nm, respectively, and 488 and 535 nm, respectively, for green fluorescence (FDA).

### 2.12. Effect of KPP-1 Endolysin on Bacterial Cells upon Oxidative Stress

Induction of oxidative stress in *K. variicola* upon KPP-1 endolysin treatment was assessed using fluorescent microscopy. Briefly, early-log phase bacterial cells at 0.18 OD_595_ and collected by centrifugation. Cells were resuspended in NB and treated with purified KPP-1 endolysin (350 µg/mL) for 4 h at 37 °C. Cells were collected by centrifugation and washed twice with PBS. To detect generation of reactive oxygen species (ROS), cells were treated with 30 µg/mL fluorescent dye 5-(and-6)-carboxy-2,7-dihydro-dichlorofluorescein diacetate (carboxy-H_2_DCFDA; Sigma-Aldrich, Saint Louis, MI, USA) from stock solution (1 mg/mL). Cells were incubated in darkness for 25 min. Cells were washed with PBS and subjected to confocal imaging using a scan head integrated with an Axiovert 200 M inverted microscope (Carl Zeiss, Jena, Germany). The excitation and emission wavelengths used for imaging were 488 and 535 nm, respectively.

### 2.13. Antibacterial Activity of KPP-1 Endolysin

The KPP-1 endolysin purified protein was tested for its ability to lyse cells of *A. hydrophila*, *A. salmonicida*, *E. coli*, *E. piscicida*, *L. monocytogenens*, *S. aureus*, *K. pneumoniae*, and *K. variicola* (indicator host) isolates. Overnight cultures of each bacterium were used to make 1% inoculums into fresh media. When the cells had grown into the early-log phase with an OD_595_ of 0.18, they were treated with 375 µg/mL purified KPP-1 endolysin. Absorbance (OD_595_) was measured at 10 h post treatment.

### 2.14. Statistical Analysis

Statistical analysis was performed using the GraphPad Prism 6.0 software (San Diego, CA, USA). One-way analysis of variance (ANOVA) with Tukey’s multiple comparison tests was performed to evaluate differences among treatment groups. A *p*-value < 0.05 was considered significant.

## 3. Results

### 3.1. Isolation, Morphology, Propagation Host, and the Host Range of KPP-1

The bacteriophage KPP-1 was isolated from the freshwater sample collected from a natural stream. The propagation host *K. variicola* isolate also had been isolated from stream water. The bacterium was confirmed to be *K. variicola* using molecular characterization based on 16S rRNA sequencing. The forward sequence generated using the 27F universal primer and 1492R universal primer was aligned together and the aligned fragment was used for the database search. The BLASTn search of the NCBI database proposed *K. variicola* strain DBBP 1 with 100% sequence identity (accessed on 25th October 2022). The sequence received 100% sequence identity against that of the *K. variicola* strain NAC12 as the second best BLASTn hit. The results obtained from the EzTaxon-e database also verified the host strain to be *K. variicola* by obtaining the best match with *K. variicola* SB5531 strain (accessed on 25th October 2022) with 100% sequence identity. Host range analysis conducted using 16S rRNA sequence confirmed 28 bacterial isolates belonging to 32 different species (Appendix A). On TA soft agar, KPP-1 produced clear plaques with distinct margins with an approximate diameter of 1–2 mm only on *K. variicola* strain (Figure 1A). The morphology of KPP-1 observed by using TEM analysis revealed an icosahedral head with approximate dimensions of 85.49 nm × 93.14 nm (width × length) and a tail structure of 101.73 nm long and 22.21 nm wide (an average dimension of five individual complete phage particles) (Figure 1B). Due to KPP-1 specificity for *K. variicola* strains, the EOI was compared against that of the *K. variicola* propagation strain and a *K. variicola* laboratory isolate. After two trials, no EOI difference was observed between both *K. variicola* isolates (Appendix A).

### 3.2. Complete Genome Sequencing and Functional Annotation of KPP-1

The complete genome of KPP-1 (GenBank accession: MT438396.1) was isolated and sequenced (Appendix A). The whole genome of bacteriophage KPP-1 consisted of 143,369 bp with 44.7% G+C content. Sequence coverage analysis revealed approximately twofold higher sequence coverage between regions of 116,316–116,829 bp, which was determined by re-aligning fastq files to the assembled genome of the KPP-1. The presence of 514 bp long direct terminal repeats (DTR) could be further confirmed by using PhageTerm software (Appendix A). This may represent the presence of direct terminal repeats similar to several reported *Klebsiella* infecting bacteriophages [50,51]. A total of 272 genes were distributed as 161 genes on the forward sequence and 111 genes on the complementary sequence. Functional analysis of each gene revealed that it consisted of 255 ORFs and 17 tRNA sequences. Among the predicted ORFs, 33 could be functionally annotated using the NCBI BLASTp algorithm, while most of the proteins were hypothetical proteins. Based on the proteomic prediction, these 33 ORFs could be classified into six categories: (1) structural, (2) nucleotide metabolism and transcription related, (3) DNA replication related, (4) packaging related, (5) lysis related, and (6) other functions (Appendix A). The BLASTp analysis did not allow identification of known antibiotic resistant genes in the KPP-1 genome, however, one or more genes may code for such a protein which must be further studied (Figure 1C).

### 3.3. Comparative Genomic Analysis and Phylogenetic Classification

BLASTn comparison of the KPP-1 genome revealed the highest sequence identity (query cover × % sequence identity), against the phages belonging to the subfamily *Vequintavirinae*. BLASTp analysis of ORFs also revealed that the majority of ORFs, 246, were present in bacteriophages belonging to the subfamily *Vequintavirinae*. According to ICTV taxonomic database [52] (accessed on 12th October 2022), the subfamily consisted of six genera, namely *Avunavirus*, *Certrevirus*, *Henunavirus*, *Mydovirus*, *Seunavirus*, and *Vequintavirus*. Among these genera, KPP-1 received the highest number of homologous ORFs in genera *Mydovirus* (Klebsiella phage vB_KpnM_KB57; 108 ORFs, vB_KpnM_BIS47; 49 ORFs, proteus phage Mydo; 42 ORFs, Mydovirus KpS8; 30 ORFs, and Klebsiella phage KNP2; 17 ORFs) (Table 1).

A comparative genomic analysis conducted using the VIRIDIC tool demonstrated that the aligned genome fraction, intergenomic similarity, and genome length ratio of KPP-1 were all in conformity with the members of the genus *Mydovirus* followed by the members of the genus *Seunavirus* which includes bacteriophages that infect *Salmonella*, *Cronobacter*, and *Escherichia* species (Figure 2).

The shared genomic synteny of KPP-1 against the members of the *Vequintavirinae* subfamily demonstrated high conservation of the genomic layout Easyfig analysis (Figure 3). Genomic cluster analysis was conducted using the CLANS software against a total of 129 whole genome sequences, which included bacteriophages infecting *Klebsiella*, *Bacillus*, *Listeria*, *Staphylococcus*, *Salmonella*, and *Escherichia* species infecting bacteriophage. The CLANS cluster analysis shows that members of the subfamily *Vequintavirinae* occur in a distinct single cluster among several other clusters produced by *Staphylococcus* Twort-like phages, *Bacillus* phages, SPO1-like phages, and other scattered *Klebsiella* phages (Figure 4).

Further, to support the phylogenetic position of KPP-1, the major capsid protein (Figure 5A) and large terminase subunit protein (Figure 5B) sequences collected from members of the *Vequintavirinae* subfamily were utilized to construct a phylogenetic tree based on the maximum likelihood method; this analysis also revealed KPP-1 as a member of the genus *Mydovirus*. Furthermore, comprehensive proteomic classification of bacteriophages can be achieved using the VipTree server, which classified KPP-1 close together with the *Adenovirus* bacteriophages, which infect members of Gammaproteobacteria (Appendix A).

### 3.4. Bacterial Growth Inhibition by KPP-1 and Its Synergy with Antibiotics

The suppression of *K. variicola* growth by KPP-1 was evaluated in vitro. When the phage particles were inoculated at 10 MOI with the host bacterium, significant suppression of bacterial growth could be observed during the first 3 h compared to that seen in the control culture. Growth suppression conferred by the KPP-1 phage was greater than the suppression produced by ampicillin (100 µg/mL). A slight increase in bacterial growth observed at 6 h post-inoculation plateaued until the 9 h post-inoculation. Moreover, bacterial growth was significantly low in cultures inoculated with KPP-1 compared to that in both the negative control and ampicillin control cultures. Beyond the 12 h period, an increase in the bacterial population was observed. At every time point, the lowest bacterial growth was evident, when cultures were co-treated with KPP-1 and ampicillin indicating a kind of synergy between the bacteriophage and the antibiotic (Figure 6).

### 3.5. Identification of the Endolysin Gene in the KPP-1 Genome

The endolysin gene of KPP-1 could be predicted using BLASTp analysis of ORFs. The 184-amino-acids-long endolysin protein had the closest homology to the glycoside hydrolase family 19 protein of *Klebsiella* phage vB_KpnM_KB57 with 100% sequence identity. Phylogenetic analysis of lytic genes gathered from related bacteriophages revealed a close phylogenetic relationship of hydrolases among the members of *Mydovirus* (Figure 7A). Assessment of the domain architecture of the KPP-1 endolysin amino acid (AA) sequence revealed a Lyz-like superfamily domain (32–177 AA) which contains the Glyco_hydro_19_catalytic domain (88–146 AA) (Figure 7B). The Glycoside hydrolase family 19 domain may be capable of breaking beta-1,4-N-acetyl-D-glucosamine linkages, the bond present in chitin structure, which can also be seen in bacterial cell walls, including in the members of the *K. pneumoniae* complex. Such modifications in lytic genes can be an adaptation of bacteriophages to facilitate efficient lysis of their host bacterium [53]. To investigate the effect of the full endolysin sequence on bacteriolysis, the complete ORF was amplified using the phage genomic DNA. The 6His-tag remained at the N-terminus of the sequence (Figure 7C). The ORF was cloned into the HisParallel 1 expression vector for protein expression and purification. The size of the purified protein based on the SDS-PAGE results was approximately 24 kDa (Figure 7D). The estimated size of the endolysin protein was 20.13 kDa with a theoretical PI value of 9.32 [54]. However, a slight increase in the size of the protein (approximately + 4 kDa) was observed due to upstream AA sequences and the 6-his tag added by the vector component.

### 3.6. The Effect of KPP-1 Endolysin on the Bacterial Cell Membrane

The effect of endolysin on bacterial lysis and survival was visualized using the PI staining method. PI is a dye that intercalates with DNA at a stoichiometric ratio of 1: 4–5 bases. The dye penetrates dead cells due to membrane permeation while not affecting live cells. Thus, PI helps to discriminate dead cells using fluorescent microscopy. The comparative study of bacteria treatment using 375 µg/mL endolysin for 10 h demonstrates the increase in cells with red fluorescence indicating the proportion of dead cells. The live cells remain green due to FDA staining. The FDA penetrates the hydrophobic cellular membranes and gets hydrolyzed by cellular esterases present only in live cells. Thus, it could effectively discriminate live cells from dead cells [55]. The negative control did not show any red fluorescence but demonstrated green fluorescence indicating that the cells were still alive (Figure 8A). This study indicates that KPP-1 endolysin was able to lyse *K. variicola* bacteria. The initiation of membrane damage triggers bacterial stress responses and increases the level of oxidative stress [56]. This effect can be due to membrane damage induced by KPP-1 endolysin. When the bacterial cells were exposed to purified endolysin an increase in oxidative stress could be observed by H_2_DCFDA fluorescence assay (Figure 8B). The antibacterial activity of KPP-1 purified endolysin was evaluated against eight bacteria isolates used in the host range analysis. Bacterial inhibition was evaluated by measuring the absorbance values at OD_595_ upon co-incubation of bacteria and KPP-1 endolysin. Efficient lysis could only be observed against *K. variicola* isolates; however, no other bacterial isolate was affected by endolysin. Treatment with 375 µg/mL KPP-1 endolysin resulted in 28.33% *K. variicola* inhibition compared to the bacterial culture alone control (Figure 8C).

## 4. Discussion

Bacteriophage KPP-1, which infects *K. variicola* was isolated from a natural freshwater stream located in Daejeon city, Republic of Korea. Host range analysis revealed that KPP-1 infects *K. variicola* but not *K. pneumoniae* or other tested bacteria, such as *Shigella*, *Enterococcus*, *Escherichia*, *Cronobacter*, *Enterobacter*, *Morganella*, *Pantoea*, *Edwardsiella*, *Aeromonas*, or *Vibrio* species. The apparent specificity and strictly lytic phenotype of KPP-1 can be successfully employed in biological control of the emerging pathogen *K. variicola* as a prominent pathogen in the *Klebsiella* complex. To ascertain these objectives, we conducted complete genome sequence analysis, phylogenetic classification, and identified endolysin as a potentially useful protein in pathogen control.

Morphological characterization of KPP-1 using the plaque assay revealed clear plaques with distinct margins with approximate diameter of 1–2 mm on the host strain. Detailed morphological analysis using TEM has revealed icosahedron head and tail structures with dimensions typical to bacteriophages of the members of *Myoviridae* according to a former ICTV classification scheme [57]. With the development of sequencing technology and classification using bioinformatics analysis, modern classification is predominantly based on sequence analysis data; hence, the KPP-1 genome sequence was assessed using several bioinformatics tools for precise identification of its phylogenetic position [58].

Whole genome sequencing of bacteriophage KPP-1 generated 4,193,426 total reads with a Q30 (%) quality-score of 95.2. The single contig generated upon de novo assembly was 143.37 Kb and it consisted of 272 total genes. Further analysis revealed that these 272 genes consist of 255 ORFs and 17 tRNA sequences. The presence of a large number of tRNA sequences can be an indication of relative independence of KPP-1 from the host protein synthesis machinery, which is favorable for its lifestyle as a strictly lytic phage. The KPP-1 genome contained two regions with approximately doubled sequence coverage indicating the potential presence of terminal repeats which is a characteristic feature of some of the bacteriophages [59]. The existence of direct terminal repeats may play a role in priming activity during DNA replication and packaging of the phage genome into newly assembled phage particles. As predicted by PhageTerm analysis, the genome packaging mechanisms of KPP-1 is similar to that of T7 type bacteriophages (Appendix A) [60].

Whole nucleotide BLAST analysis revealed that the KPP-1 phage is closely related to the *Klebsiella* phage vB_KpnM_KB57 with 90.2% sequence identity. It is a phage belonging to the subfamily *Vequintavirinae* of the class Caudoviricetes. Functional assignment for each predicted ORF was performed using BLASTp analysis of the corresponding AA sequence. Out of a total of 255 ORFs, only 33 ORFs (12.94%) could be related to proteins available in the NCBI database while the majority of proteins were hypothetical proteins whose specific functions are yet to be elucidated. Out of the total number of ORFs, 48.64% were found to be significantly similar to those of the *Klebsiella* phage vB_KpnM_KB57, 22.07% were found to be similar to corresponding proteins in the *Klebsiella* phage vB_KpnM_BIS47m, 18.9% were found to be similar to those of Proteus phage Mydo, 13.51% were similar to those of *Mydovirus* KpS8, and 7.6% to those of the *Klebsiella* phage KNP2. All of these are members of the genus *Mydovirus* of the subfamily *Vequintavirinae*. To further elaborate on phylogenetic comparison of the KPP-1 genome, we downloaded taxonomic data on all bacteriophage species listed in the ICTV taxonomic browser belonging to subfamily *Vequintavirinae* and examined it using VIRIDIC analysis. The KPP-1 genome sequence was grouped among that of other members of the genus *Mydovirus* (Figure 1). The second closest relatives of KPP-1 belonged to the genus *Seunavirus*. To visualize sequence homology between KPP-1 and the members of the subfamily *Vequintavirinae* we conducted Easyfig analysis by selecting members belonging to *Mydovirus*, *Seunavirus*, *Henunavirus*, and *Avunavirus* bacteriophages. KPP-1 showed the closest genomic synteny with the members of *Mydovirus* and was distantly related to the genus *Avunavirus* (Figure 3). To obtain a broader picture of the relative phylogenetic position of KPP-1, we conducted a CLANS cluster analysis using the nucleotide sequences of 129 phages belonging to *Klebsiella*, *Listeria*, *Bacillus*, *Staphylococcus*, and *Escherichia* phage species. A BLAST analysis generated a cluster arrangement which revealed that all *Vequintavirinae* members are arranged in a single cluster, while the rest of the *Klebsiella* phage taxids remained scattered as singletons. Selected *Listeria*, *Bacillus*, *Staphylococcus*, and *Escherichia* phages also generated several distinct clusters, such as Twort-like phages, BCP8-like phages, and SPO1-like phages. Furthermore, results of phylogenetic tree construction using the major capsid protein and large terminase subunit were in conformity with KPP-1 as a member of the genus *Mydovirus*. According to ICTV classification guidelines, for a phage to be considered as a novel phage, there should be a minimum 5% difference in nucleotide sequence identity against the available members of the respective classification. Based on the present criteria, KPP-1 can be proposed as a novel member of the genus *Mydovirus* due to significant sequence difference (> 5%) against the closest member of the genus.

The use of strictly lytic phages for pathogen control has been successfully demonstrated. Owing to their environmentally friendly nature, host specificity, and the absence of difficulties such as the development of antibiotic resistance, bacteriophages gained special attention in pathogen control [61]. In vitro infections with KPP-1 against *K. variicola* not only revealed that it can effectively suppress the bacterium for as long as 12 h, but also showed that it can enhance the functions of antibiotics upon co-incubation. Bacteriophages are not affected by antibiotics; therefore, we could observe a synergistic effect of bacterial growth suppression between KPP-1 and ampicillin. Ampicillin, as a member of the antibiotic β-lactamase class, interferes with the biosynthesis of bacterial cell walls [62]. Therefore, weakened cell walls may facilitate phage entry, which may be the reason for the synergistic effect in *K. variicola* suppression. However, this phenomenon requires further investigation.

Every bacteriophage has biological machinery to lyse the host bacterium. Therefore, the identification of genes involved in host lysis may provide clues and genomic resources necessary for pathogen control [63]. In our search for lysis-related genes in the KPP-1 genome, we recognized gene locus KPP_11591 as a potential gene encoding the enzyme responsible for bacterial lysis. The ORF thus recognized was homologous to other hydrolases found in members of the genus *Mydovirus;* however, it had never been tested for lysis of the host in the protein form. In this study, we cloned the particular endolysin ORF, expressed it, and assessed the ability of the protein to lyse bacteria and to control *K. variicola*. Upon treating bacteria with 375 µg/mL KPP-1 endolysin, significant lysis of the host was observed. The effect of KPP-1 endolysin on bacterial viability could be visualized quantitatively using PI staining and confocal microscopy examination. Compared to the negative control, KPP-1 endolysin increased bacterial cell permeability, causing PI to intercalate with bacterial DNA, while FDA stained only the live cells. Furthermore, an increase in oxidative stress in bacterial cells may occur because of the cellular damage caused by cell permeation due to endolysin activity. We also observed that the *K. variicola* inhibition by purified endolysin treatment was not as potent as that by the KPP-1 bacteriophage. A possible explanation might be the involvement of several other proteins in the lysis process, such as the holin protein in some other bacteriophages. Holin creates minute punctures in the cell wall that increase the accessibility for endolysin to reach the cell wall peptidoglycan structure [64]. Unfortunately, a protein with a similar function to holin could not be discovered using bioinformatics prediction tools due to the lack of curated candidate proteins in the NCBI database. Therefore, further studies may be required to optimize the function of endolysin. Furthermore, considering the strict specificity of KPP-1 towards *K. variicola* strains, the phage may provide a unique avenue to distinguish *K. variicola* from its close relative *K. pneumoniae*, which are often mistaken for the other. Such a phage typing approach requires further elaborated studies to identify the depth and breadth of host recognition using a large library of members belonging to the *K. pneumoniae* complex.

## 5. Conclusions

The present investigation reveals bacteriophage KPP-1 as a strictly lytic bacteriophage against the multidrug-resistant *K. variicola* strain. The strictly lytic nature of KPP-1 and the availability of a plethora of useful genes such as endolysin may be important for future pathogen control endeavors. Furthermore, this study allows us to speculate that bacteriophages can affect antibiotic function, and they might augment the role of antibiotics, which requires further elucidation of the mechanisms. Due to its genetic distance to existing bacteriophages, we propose that KPP-1 could be a novel species of the genus *Mydovirus* of the subfamily *Vequintavirinae*. To date, this is the only bacteriophage deposited in the NCBI database that is reported to infect a *K. variicola* strain. Characterization of KPP-1 may be useful for future phage classification studies with the discovery of similar bacteriophages with useful properties.

## Figures and Tables

**Figure 1 microorganisms-11-00207-f001:**
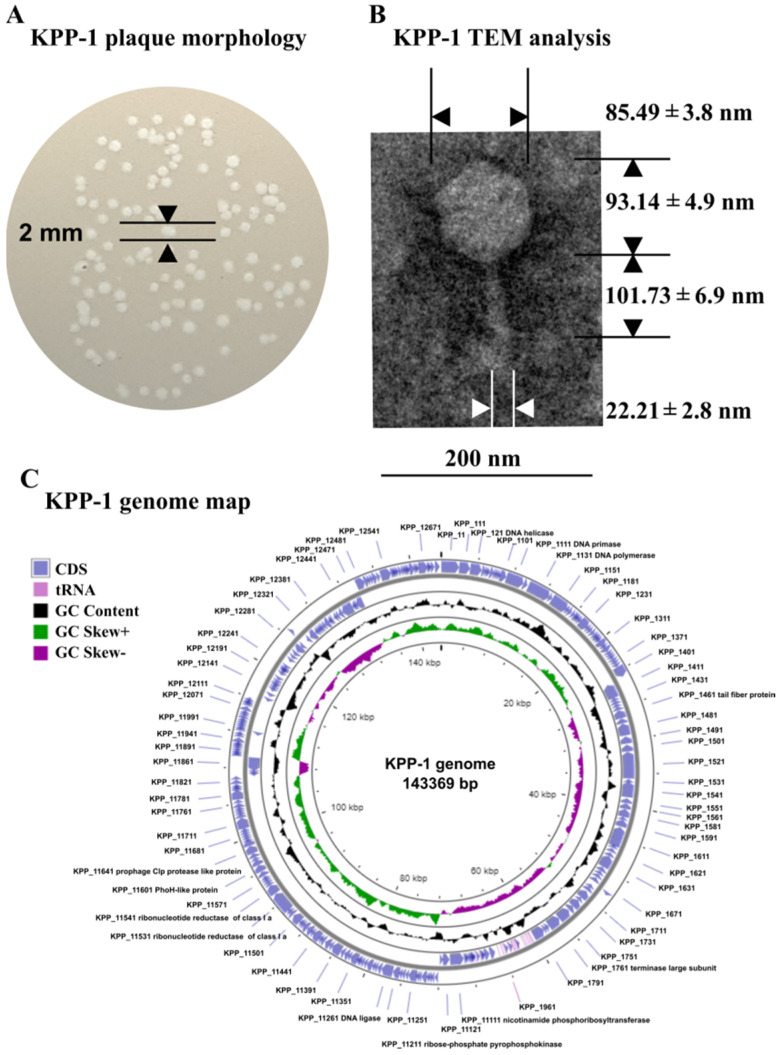
Morphological characterization and genome map of KPP-1. (**A**) KPP-1 creates clear plaques on lawns of *K. variicola* with an approximate 1–2 mm diameter. (**B**) TEM analysis of KPP-1 reveals icosahedral head and tail structures. Approximate dimensions are given in the figure (magnification: 100,000×). (**C**) Genomic map of KPP-1 demonstrates CDS, tRNA, GC content, GC skew+, and GC skew—graphs on the diagram. CDS; coding sequences.

**Figure 2 microorganisms-11-00207-f002:**
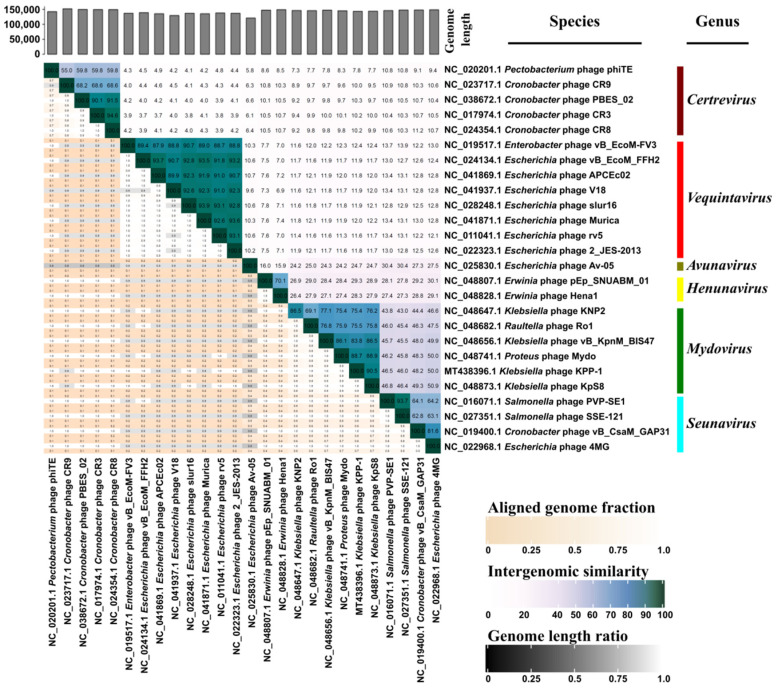
VIRIDIC analysis. Twenty six bacteriophage genomes (ICTV classified) belonging to subfamily *Vequintavirinae* were compared. The diagram provides information about aligned genome fraction, intergenomic similarity, and genome length ratio of selected genomes compared to each other. The genus-level classification was also demonstrated.

**Figure 3 microorganisms-11-00207-f003:**
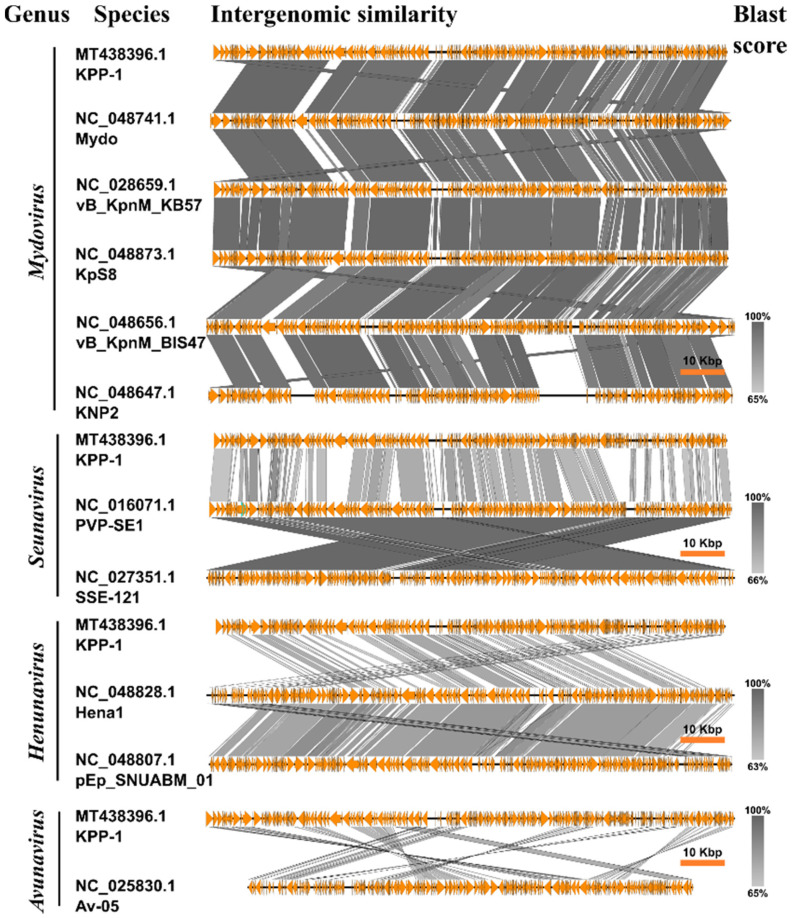
Easyfig analysis. Shared genome synteny of KPP-1 was compared against close relatives of KPP-1. The highest genomic synteny of KPP-1 is seen against members of the subfamily *Vequintavirinae*.

**Figure 4 microorganisms-11-00207-f004:**
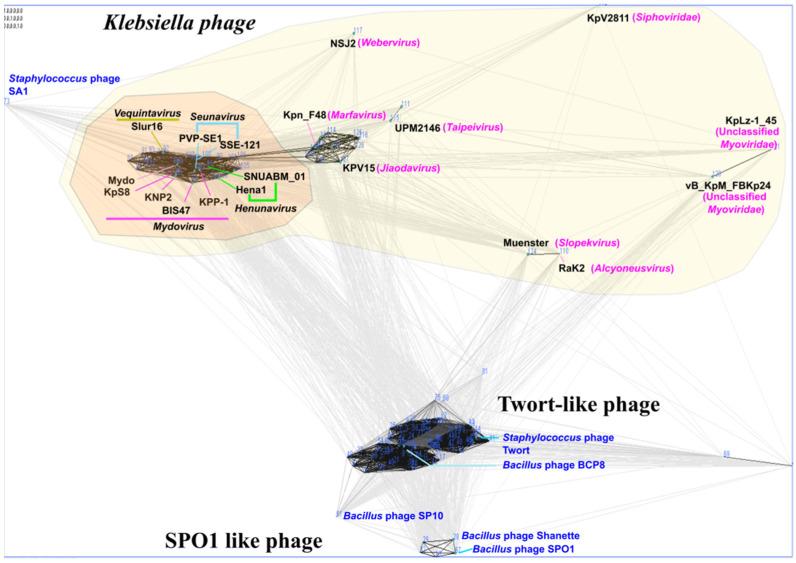
CLANS cluster analysis. Cluster analysis was conducted using genomes of 129 bacteriophage that contained bacteriophages infecting *K. pneumoniae*, *Listeria* spp., *Bacillus* spp., *Staphylococcus* spp., *Salmonella* spp., and *Escherichia* spp. The intensity of and the distance of each line demonstrates the strength of the BLAST score comparing each of the prescribed genomes. Highlighted regions indicate all *Klebsiella* phage taxids available in the NCBI database.

**Figure 5 microorganisms-11-00207-f005:**
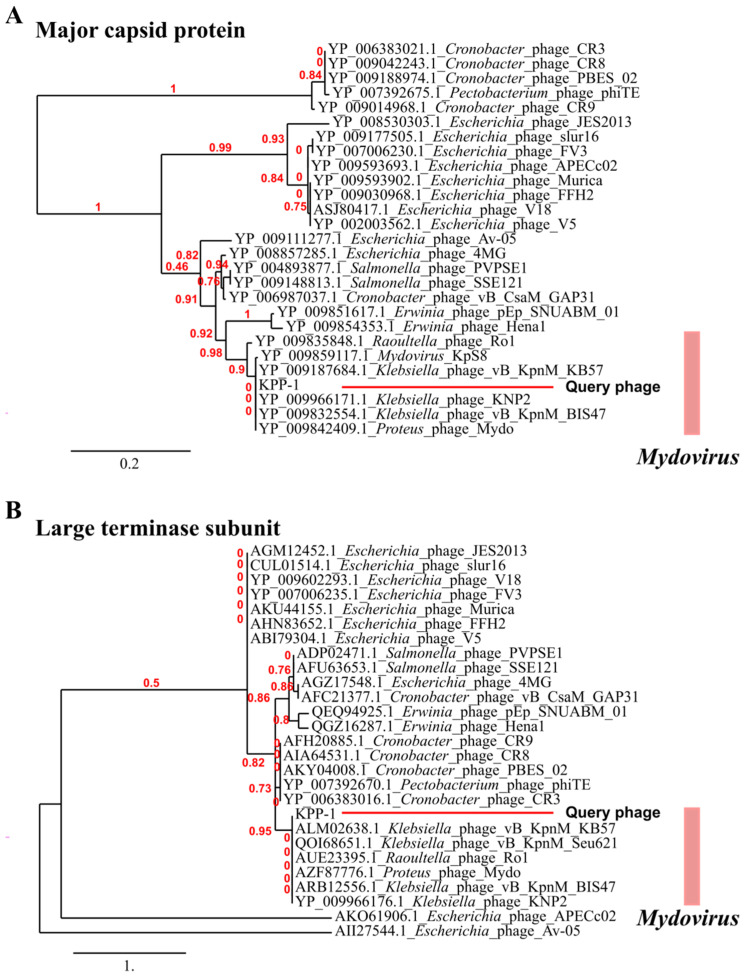
KPP-1 phylogenetic tree constructed using major capsid protein and the large terminase subunit. (**A**) phylogenetic tree based on major capsid protein. (**B**) phylogenetic tree based on large terminase subunit protein. Phylogenetic trees were constructed using maximum likelihood method for relevant sequences collected from bacteriophages belong to sub family *Vequintavirinae*. KPP-1 was grouped among the members of the genus *Mydovirus*.

**Figure 6 microorganisms-11-00207-f006:**
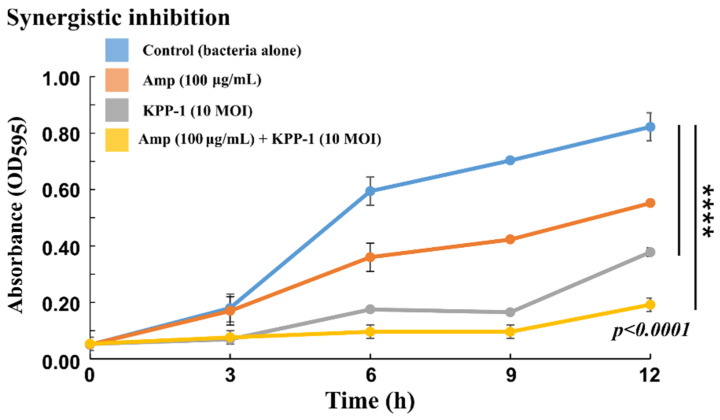
Synergy of *K. variicola* growth suppression. Synergistic suppression of *K. variicola* upon co-incubation with KPP-1 and ampicillin is demonstrated. The lowest bacterial growth was observed when both KPP-1 and ampicillin were used in combination. The suppression of KPP-1 at the 10 MOI level was shown to be more potent than ampicillin alone. **** indicates significant difference between antibiotic and phage-free control. The significant difference was determined at *p* < 0.05.

**Figure 7 microorganisms-11-00207-f007:**
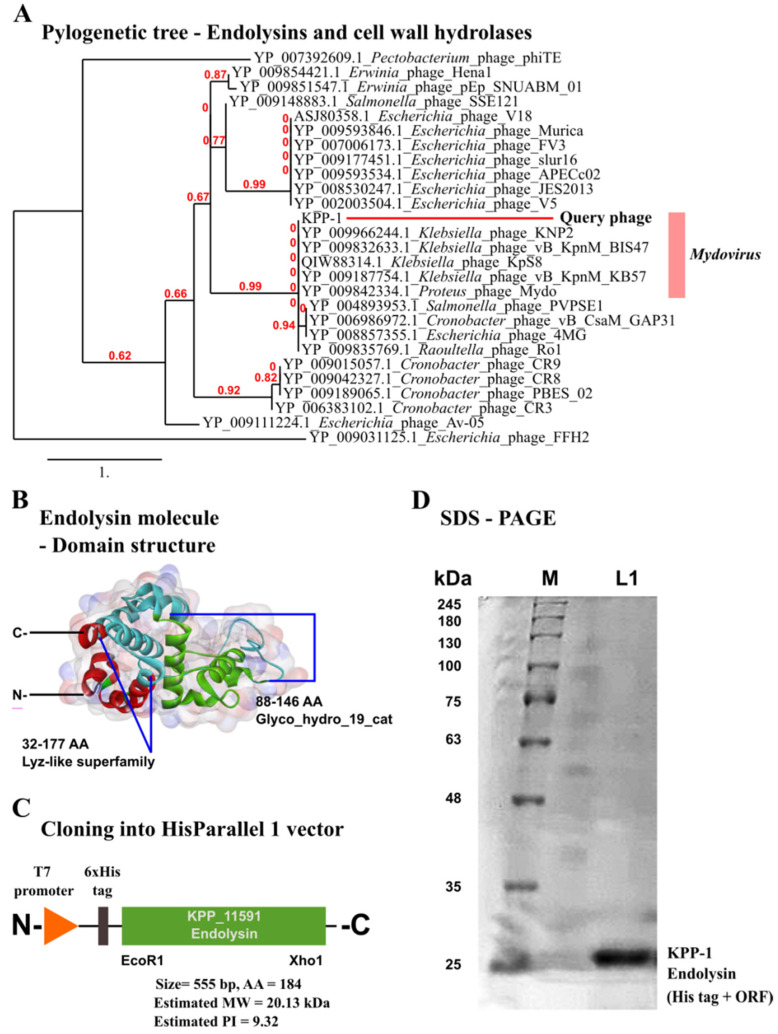
Phylogenetic comparison, distinct domains, cloning, and purification of KPP-1 endolysin. (**A**) Phylogenetic tree was constructed using maximum likelihood method for endolysin and cell wall hydrolase proteins. The query KPP-1 endolysin (Locus tag: KPP_11591) was highlighted. (**B**) Identification of conserved domains in KPP-1 endolysin demonstrates Lyz-like superfamily (32–177 AA) and Glyco_hydro_19_cat catalytic domain (88–146 AA). (**C**) Cloning of KPP-1 endolysin ORF into His-Parallel 1 vector was achieved using EcoR1 and XhoI restriction enzyme digestions. (**D**) SDS-PAGE (12%) of purified KPP-1 endolysin protein with an approximate size of 24 kDa.

**Figure 8 microorganisms-11-00207-f008:**
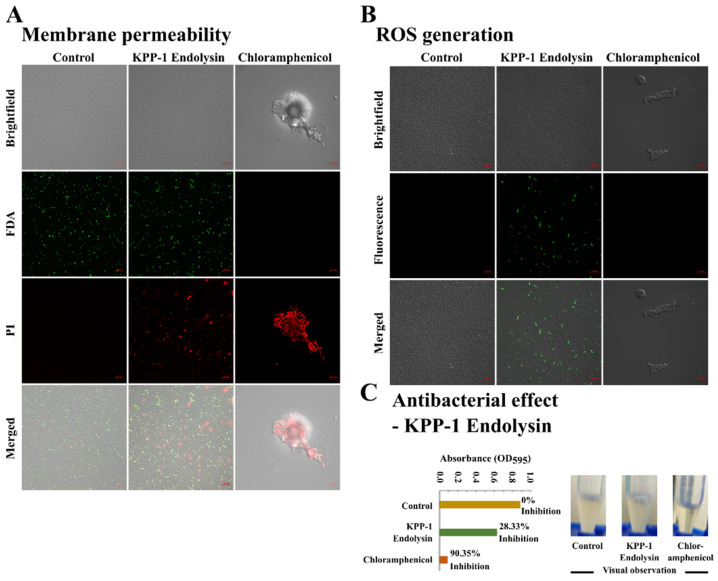
Assessment of membrane permeability, oxidative stress, and lytic ability of KPP-1 endolysin. (**A**) KPP-1 endolysin-induced membrane permeability of *K. variicola* was assessed using PI staining method. Fluorescent confocal microscopy analysis was used to compare treatment differences among PBS control, KPP-1 endolysin (375 µg/mL) treated, and the chloramphenicol (35 µg/mL) treated groups. Live and dead cell discrimination was based on FDA; green fluorescence and PI; red fluorescence, respectively. (**B**) KPP-1-induced oxidative stress in *K. variicola* was assessed by detecting ROS generation. The ROS generation was detected by H_2_DCFDA staining and the emission of green fluorescence signal was examined by confocal microscopy analysis. The scale bar indicates a 10 µm distance. (**C**) Antibacterial activity of KPP-1 purified endolysin against *K. variicola* isolate (propagation host). Bacterial inhibition was quantitatively evaluated by measuring absorbance at OD_595_. Treatments used were, control (bacteria only), KPP-1 endolysin (bacteria and KPP-1 endolysin co-incubation), and chloramphenicol (bacteria and chloramphenicol co-inoculation; positive control).

**Table 1 microorganisms-11-00207-t001:** Genomic and proteomic features of ICTV-classified *Vequintavirinae* phages and KPP-1.

Bacteriophage	Genbank Accession No.	Genome Length (Kb)	G+C mol%	No. of CDS	No. of tRNAs	Genes	Pseudogenes	Query Cover (A)	% Identity (B)	Sequence Identity (A × B)
Klebsiella phage KPP-1	MT438396.1	143.37	44.7	255	17	272	1	100	100.00	100.00
Klebsiella phage vB_KpnM_KB57	NC_028659.1	142.99	44.6	245	16	261	1	92	98.06	90.21
Klebsiella phage vB_KpnM_BIS47	NC_048656.1	147.44	44.6	262	0	262	0	84	96.16	80.77
Proteus phage Mydo	NC_048741.1	145.13	44.8	264	23	287	0	89	98.80	87.93
Mydovirus KpS8	NC_048873.1	143.8	44.6	261	17	278	0	91	99.55	90.59
Klebsiella phage KNP2	NC_048647.1	146.2	45.2	211	0	211	0	76	98.62	74.95
Salmonella phage PVPSE1	NC_016071.1	145.96	45.6	244	24	268	0	25	77.73	19.43
Salmonella phage SSE121	NC_027351.1	147.74	45.3	242	0	242	0	26	77.73	20.20
Erwinia phage Hena1	NC_048828.1	148.84	48.4	240	26	266	0	8	78.01	6.24
Erwinia phage pEp_SNUABM_01	NC_048807.1	147.32	48.7	249	26	275	0	8	80.38	6.43

## Data Availability

The annotated complete genome sequence of the KPP-1 phage reported herein is available at GenBank under accession number MT438396.1. Data will be made available on request.

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
