# Peer review of "Genome Characterization of Bacteriophage KPP-1, a Novel Member in the Subfamily Vequintavirinae, and Use of Its Endolysin for the Lysis of Multidrug-Resistant Klebsiella variicola In Vitro"

_microorganisms, 2023, doi:10.3390/microorganisms11010207_

Round 1

Reviewer 1 Report

line 16: Change "The bacteriophage KPP-1 was strict lytic on K. variicola" to "The bacteriophage KPP-1 was strictly lytic on K. variicola"

line 22: Change "the closest neighborhood can be found" to "the closest neighbour can be found"

line 41: Change "Though it is considered an emerging" to "It is now being considered an emerging pathogen, with plenty of reports..."

line 46: Change "There has been" to "Additionally, there is evidence..."

line 47: Change "the clinical landscape of the K. variicola is revealing at" to "the clinical landscape of the K. variicola is changing at"

line 49: Change "It can also be observed that K. variicola infections are more" to "It is also known that K. variicola infections are more"

line 67: Change "and about 23 annotated Klebsiella phage" to "and 23 annotated Klebsiella-infecting phages"

line 69: Change "These 23 Klebsiella phages were belonging to 13 Genera" to "These 23 Klebsiella phages belong to 13 Genera"

line 83: Change "we introduce yet another novel bacteriophage" to "we introduce a novel bacteriophage"

line 87: Change "member of the Mydovirus genus, of the Subfamily" to "member of the genus Mydovirus, of the subfamily"

There are many instances like the ones I've highlighted above, so please address this throughout the manuscript.

line 101: Change "Cronobacter sakazakii Enterococcus faecalis, Escherichia coli," to "Cronobacter sakazakii, Enterococcus faecalis, Escherichia coli,"

line 110: Change "sequencing 16 rRNA using" to "sequencing 16S rRNA using"

line 123: Change "Herein, 100 μL of decimally diluted samples in SM buffer" to "Following ten-fold serial dilution of sample in SM buffer, 100 μL of each dilution was mixed..."

line 128: Change "and sent for incubation at 37 oC for 12 h" to "and incubated at 37 oC for 12 h"

line 130: How was the filter sterilized using filters..?

line 140: Change "Host range analysis were conducted by plaque-forming assay" to "Host range analysis was conducted by plaque assay"

Please add a reference to Supplementary Table 1 in section 2.4.

In the figure legend of Supplementary Figure 3: Change "VipTree web severer" to "VipTree web server" and italicize Mydovirus

line 168: Change "Trimmomatics" to "Trimmomatic"

line 185: Change "The whole genome sequence was compared against the NCBI BLASTn algorithm to recognize the closest phylogenetic proximity to the available phage genomes" to "The genome nucleotide sequence was compared with the NCBInr database using the BLASTn algorithm to  identify the closest relative among available phage genomes"

The IMG/VR database is far better to search against.

line203: Change "were set as media alone" to "were set as medium alone"

line 210: Change "reaction (PCR) using genomic DNA" to "reaction (PCR) using phage genomic DNA"

line 249: Change "washed with PBS twice times" to "washed twice with PBS times"

line 274: Change "on 16 sRNA sequencing" to "on 16S rRNA sequencing" - Check this throughout the paper.

Please move the Genbank accession number to section 3.2.

Figure 1 legend: Change "an approximate 1-2 mm of diameter" to "an approximate 1-2 mm diameter"

Figure 1 legend: Change "Approximate dimensions are mentioned in the figure. Magnification: 100, 000x." to "Approximate dimensions are given in the figure (Magnification 100000 X)"

line 307: Please stay consistent with formatting "positions at 116,316 and 116829 (Supplementary figure 2)."

The authors state in line 316 that "There were no genes associated to the integration of antibiotic resistance". With 87% of the genome (222/255*100) encoding hypothetical proteins, I think the authors should consider re-wording their statement to indicate that they could not identify known antibiotic resistance genes on the genome, however, one or more of the hypothetical proteins may code for such a resistance gene.

Figure 2 legend line 320: Change "The query phage KPP-1 is 320 underlined in red color." to "The query phage KPP-1 is 320 underlined in red."

line 326: Change "identity; the query cover × % sequence identity, against" to "identity (the query cover × % sequence identity) against"

line 334-335: Delete this sentence "The feature similarity of these bacteriophages is presented in the Table. 1." and add "(Table 1)" at the end of the previous sentence ending with "KNP2; 17 ORFs)." in other words "KNP2; 17 ORFs) (Table 1)."

line 337 and 338: Italicize the virus genus names.

line 340: Change "The shared genomic synteny of 339 KPP-1 against the members of the Vequintavirinae subfamily is displayed in figure 3, which 340 demonstrates the results of Easyfig analysis (Figure 3)." to "The shared genomic synteny of 339 KPP-1 against the members of the Vequintavirinae subfamily demonstrates high conservation of the genomic layout (Figure 3)."

The authors state in line 344-346 that "The sequence diversity demonstrated by Klebsiella phages was higher than conserved bacteriophages that were present in Bacillus and Staphylococcus phage clusters (Figure 4).", however provide no network metrics (Betweenness, closeness, degree correlations etc.) to support this.

The authors keep stating which software programs were used in the construction of various analyses in the Results sections which were already mentioned in the Methods section. For example in line 348-350: "were utilized to construct the Maximum Likelihood Method based phylogenetic tree utilizing Phylogeny.fr program using One Click method with G-blocks activated." Please refrain from doing this. Also remove any mention of these in the Figure legends.

In the text the authors state that the theoretical size of the endolysin is ~20kDa, then in the Figure 7 legend they  sate that the protein is 25kDa. Can the authors please clarify which of these is correct. It is possible that a 20kDa protein could've run at a slightly higher molecular weight on the PAGE gel. If this is the interpretation, then please state that.

I would suggest to combine sections 3.6, 3.7 and 3.8.

The authors mention a "rapid lysis profile" in line 468. It may have been useful to include a one-step growth curve to demonstrate this point.

A large portion of the "Discussion" is merely a repeat of the results section rather than thoughtful discussion of what their results mean in light of what is already known about these viruses. I suggest that the authors make an attempt to reduce or eliminate as far as possible stating their results in the Discussion section a second time.

line 513: Change "The use of strictly lytic phages for pathogen control has been successfully employed 513 in pathogen control." to "The use of strictly lytic phages for pathogen control has been successfully employed."

line 544: Change "function to Holing could not" to "function to holin could not"

I have by no means highlighted every error in the manuscript. I urge the authors to go through their manuscript with care and correct small editorial errors and language errors before re-submission.

Reviewer 2 Report

The authors present a novel bacteriophage isolated from a freshwater sample. They go ahead and perform a plethora of experiments to support their claim. The analyses may have been done properly but there are some serious flaws that need to be corrected.

First of all, there is a need for extensive editing, particularly in the grammar. Secondly, the images are of low resolution and better images need to be created. Moving on, I am missing crucial information in many parts in the Materials and Methods. In particular, where was the fresh water stream and  where were the organisms mentioned in 2.1 isolated from? Which versions of the programs were used? What were the parameters? Why did they authors choose so many different ORF prediction pipelines when in the end only a handful of ORFs were given a general function? How did the inclusion of other programs imporve the PROKKA output? My concern on the programs continues in the rest of Materials and Methods. There is a protein model presented on page 15 but nothing has been stated about how this was produced. What is the significance of the genome map (figure 1)? If the authors want to keep it, it also needs to be rendered in higher resolution. On page 10 why do we see the comparison between KPP-1 and non-Mydovirus phage genomes? How is that different from what we see in figure 2 in relation to where the phage belongs? To that end, if the genomes were slightly repositioned, then the syntenic blocks would show almost on top of each other instead of slightly titled, but that's a minor comment. Considering how bacteriophages are very explicit to their host, why did the authors feel the need to perform the CLANS analysis including other genomes (Accession numbers missing) of bacteriophages that do not infect Klebsiella variicola?

My expertise does not allow me to comment on the in vitro experiments and I hope the editors will find a reviewer capable of reviewing that part. To that end, the title of "Genome characterization..." seems to be either misleading or incomplete considering that most of the effort done here was on the growth experiments and endolysin purification.

Reviewer 3 Report

Dear authors,

I have read with interest your manuscript on “Genome characterization of bacteriophage KPP-1, a novel member in the subfamily Vequintavirinae for the control of multi-drug-resistant Klebsiella variicola”. The authors have performed the bioinformatical analysis in detail and assessed the KPP-1-encoded endolysin. Their analysis methods seem to be appropriate for this type of study.

Some specific queries are listed below;

In line 168, the authors described prokka tool was used for genome assembly. But as far as I know, prokka is an annotation tool but not an assembly tool. Please cite other appropriate tools.

In Fig. 2, the numbers in the heatmap are hard to read.

In Fig. 4, “Staphylocossus phage” should be “Staphylococcus phage”

In Fig. 7B, how did you determine the structure of KPP-1 endolysin?

In line 402-403, how did you detect the two domains (Lys-like superfamily domain and Glyco_hydro_19_catalytic domain)? Did you use any domain database such as Pfam? Please clarify this point. In addition, in line 543-545, the authors mention that Holin protein could not be detected. I recommend the phage protein database “phrog” (https://phrogs.lmge.uca.fr) or pfam (http://pfam.xfam.org) to find the protein involved in Holin activity.

If you discover the holin gene, I would recommend purifying the protein as well as endolysin and carrying out the experiments done in Fig. 8. Figure 8 is not shown in the text.

Reviewer 4 Report

Phage therapy is promising due to bacterial drug resistance. In this study, the authors isolated a phage of Klebsiella variicola. The general biological features and genomic properties were analyzed in this manuscript, including endolysin protein. The content of manuscript is comprehensive. The major question is that the classification of phages is refereeing to an old method. The new classification should be used in the manuscript(A Roadmap for Genome-Based Phage Taxonomy, Virus, 2021, 13, 506.). Other questions are below.

1.     In Figure 1B a high-resolution figure is better to replace the old one.

2.     The classification of phages refers to the new classification in Figures 2, 3,4 and 5.

3.     The genus should be italic in Figures 5 and 7A.

Round 2

Reviewer 3 Report

I agree that you have overall improved the manuscript substantially. The revised manuscript is suitable for publication.

Author Response

We thankful to the reviewer for reviewing the manuscript.